# Effects of Commercial Probiotics on Colonic Sensitivity after Acute Mucosal Irritation

**DOI:** 10.3390/ijerph19116485

**Published:** 2022-05-26

**Authors:** Laura López-Gómez, Jaime Antón, Yolanda López-Tofiño, Bianca Pomana, José A. Uranga, Raquel Abalo

**Affiliations:** 1Department of Basic Health Sciences, University Rey Juan Carlos (URJC), 28922 Alcorcón, Spain; laura.lopez.gomez@urjc.es (L.L.-G.); jaimeanton.b@gmail.com (J.A.); yolanda.lopez@urjc.es (Y.L.-T.); biancapomana.bp@gmail.com (B.P.); 2High Performance Research Group in Physiopathology and Pharmacology of the Digestive System (NeuGut-URJC), University Rey Juan Carlos, 28922 Alcorcón, Spain; 3Associated I+D+i Unit to the Institute of Medicinal Chemistry (IQM), Scientific Research Superior Council (CSIC), 28006 Madrid, Spain; 4Working Group of Basic Sciences in Pain and Analgesia of the Spanish Society of Pain, 28046 Madrid, Spain

**Keywords:** animal model, inflammatory bowel disease, irritable bowel syndrome, probiotics, visceral pain

## Abstract

Gastrointestinal pathologies associated with abdominal pain, such as irritable bowel syndrome or inflammatory bowel disease, lack sufficiently effective treatments. In our study we have used a rat model of visceral pain (72 animals; *n* = 8–13 per experimental group) to analyze the consequences of intracolonic administration of the irritant acetic acid on visceral sensitivity, histology of the colonic wall, and inflammatory response. Moreover, we have studied the possible beneficial effects of a pretreatment with a commercial probiotic (Actimel^®^). Contrary to expectations, acetic acid application (7 cm proximal to the anus) decreased the nociceptive response to intracolonic mechanical stimulation, with a slight increase in the histological damage of colonic mucosa. The intensity of these changes depended on the concentration (4% or 0.6%) and the time of application (30 or 60 min). Pretreatment with probiotics (by daily gavage, for 1 week) normalized the values obtained in the visceral sensitivity test but revealed an increase in the number of macrophages. These results suggest a possible activation of inhibitory mechanisms early after colonic irritation, not previously described (which need further experimental confirmation), and the ability of probiotics to normalize the effects of acetic acid. In addition, pretreatment with probiotics has a direct effect on immune functions, stimulating macrophagic activity.

## 1. Introduction

Inflammatory bowel disease (IBD) and irritable bowel syndrome (IBS) are two of the most common gastrointestinal (GI) disorders in the world. The etiopathogenesis of these diseases is not yet fully clarified. The evidence shows that they are multifactorial diseases influenced by genetic and environmental factors (in particular, those that produce psychological stress), with activation of the immune response (greater in IBD) and an increase in the permeability of the epithelial barrier, anomalies in the microbiome, and in the perception of pain [1,2]. Likewise, comorbidity has been observed with psychological disorders, such as depression and anxiety [3,4]. Similarly, visceral sensitivity is increased, with development of allodynia (pain responses to non-painful stimuli) and hyperalgesia (exaggerated pain responses to painful stimuli), manifested as severe and recurrent abdominal pain [5].

Unfortunately, the pharmacological treatments currently used are not completely effective or safe to improve IBS and IBD [6,7]. New strategies are being studied to improve the quality of life of patients. In this context, diet-related strategies could be particularly useful for treating different aspects of IBS and IBD, including the increased visceral sensitivity [8,9,10].

Among these strategies, the modification of the microbiota through the administration of probiotics is one of the most promising possibilities. Probiotics are live microorganisms that, when administered in adequate amounts, confer a health benefit on the host [11]. The gut microbiota participates in different physiological functions, such as intestinal development, processing and digestion of nutrients, development of immune responses, resistance to pathogens, and control of lipid metabolism, among others. Intestinal dysbiosis (changes in the composition and number of bacterial populations within the gut) has been observed in patients with IBS and IBD [2,12,13]. For this reason, the controlled alteration of the microbial populations of the gut microbiota through probiotics has emerged as a possible strategy to prevent some symptoms of these pathologies. Indeed, in recent years, positive and promising results have been observed regarding various symptoms: reduction of inflammation, reduction of abdominal pain and visceral hypersensitivity, prevention of negative effects of antibiotic treatments, and improvements in bowel habits and quality of life of patients [14,15,16]. However, more studies are required to clarify which may be the most effective and safe protocols for administration and the underlying mechanisms.

Actimel^®^ is a commercial preparation of probiotics widely consumed. It is available in supermarkets and local shops, and every day a large number of people take this product. However, it has not been investigated in depth whether its consumption may be relevant in IBS or IBD and the visceral pain associated with these illnesses.

Thus, our aims were (1) to analyze, in rats, the consequences of intracolonic administration of the nonspecific chemical irritant acetic acid on visceral sensitivity, as well as on the structure of the colonic wall and the number of immunocytes, and (2) to assess the effect of pretreatment with Actimel^®^ on these parameters.

## 2. Materials and Methods

The experiments were carried out in February–June 2019 at Universidad Rey Juan Carlos (Madrid, Spain) and were designed and performed according to the EU Directive for the Protection of Animals Used for Scientific Purpose (2010/63/EU) and Spanish regulations (Law 32/2007, RD 53/2013 and order ECC/566/2015) and approved by the Ethic Committee at Universidad Rey Juan Carlos (URJC) and Comunidad Autónoma de Madrid (PROEX 063/18, PROEX 023/19). Every effort was made to minimize animal pain and discomfort as well as to reduce the number of animals used.

### 2.1. Animals

Seventy-two male young adult Wistar rats (2–3 months old, 300 ± 37 g body weight; *n* = 8–13 per experimental group) were obtained from the Veterinary Unit of URJC and randomly housed in standard transparent cages (60 × 40 × 20 cm, 2–4 animals/cage) at a constant temperature (20 °C) and relative humidity (60%) and maintained under a 12 h light/dark cycle (lights turned on at 8 a.m.) with free access to chow pellets (Harlan Laboratories Inc., Indianapolis, IN, USA) and sterile tap water.

### 2.2. Experimental Protocol

The study was divided into two phases (Figure 1). In both, functional (visceral sensitivity) and structural (colonic wall histology and immunohistochemistry) effects were evaluated by trained researchers, blinded to the treatment received by the animals.

During phase I, the effects of acetic acid on visceral sensitivity and structure of the colonic wall were studied, and the most suitable experimental model to be used in phase II was chosen. For this, the animals were administered intracolonic saline (control, SAL) or acetic acid (ACE) at two different concentrations: 0.6%, which should cause sensitization of afferent fibers, without concomitant inflammation (IBS model); and at 4%, which should cause sensitization of afferent fibers associated with an inflammatory response (IBD model). Visceral sensitivity was evaluated at two different time points after administration: 30 (SAL30/ACE30) or 60 min (SAL60/ACE60). Thus, the study was focused on the early stages of the models, occurring soon after irritation.

During phase II, the effects of a previous administration of probiotics were evaluated on the experimental model selected during phase I. For this, the selected groups were administered 1 mL of Actimel^®^ (ACT) using an orogastric probe once a day during the 7 days prior to the visceral sensitivity test. The last ACT administration was carried out 30 min before the test. The composition of Actimel^®^ is provided in Table A1, Appendix A.

### 2.3. Visceral Sensitivity

The evaluation of visceral sensitivity was carried out following the methodology described in previous works of our research group [17,18]. Firstly, the animals were sedated using Sedator^®^ (medetomidine hydrochloride, 1 mg/kg, ip). Then, a 10 cm longitudinal line was drawn over the linea alba of the abdomen. Transverse lines were drawn every 2 cm to better visualize the contractions during the recordings. Next, fecal material was gently removed from the rectum. For intracolonic administration, animals were placed vertically upside down and, using a rounded tip metal probe, 0.7 mL of the corresponding compound (SAL or ACE) were slowly introduced at a depth of 7 cm, for 2 min, to ensure a homogeneous and appropriate administration. Animals were wrapped in a cloth to reduce heat loss and avoid hypothermia for 30 or 60 min, depending on the experimental group. Then, a 5 cm long latex balloon lubricated with Vaseline was inserted through the anus into the colon 7 cm inside the colorectum. The catheter to which the balloon was connected was fixed to the tail of the rat with Parafilm^®^, to avoid its expulsion.

Sedation was reverted with Revertor^®^ (atipamezole hydrochloride, 0.66 mg/kg, ip). After waking up, the rat behavior was recorded using a video camera (iPad; Apple, Madrid, Spain) located 30 cm below the recording cage floor. The first 5 min were only used to confirm the normal behavior of the rat after recovery from sedation and were discarded; thereafter, the pressure of the intracolonic balloon was gradually increased, using a sphygmomanometer. A tonic stimulation protocol was used, where pressure was increased from 0 to 80 mm Hg, in steps of 20 mm Hg every 5 min, to finally return to 0 mm Hg again; for each pressure value, a single stimulus was applied and maintained for 5 min.

The videos were exported as series of frames (1/s), using Quick Time Player Pro for Windows (v.7.7.4; Apple Inc., CA, USA). Visceral sensitivity was measured as abdominal contractions in response to mechanical intracolonic stimulation. An abdominal contraction was considered as a depression of the abdomen where transverse lines approached one another. Thus, each frame was analyzed to determine if the rat abdomen was contracted or relaxed. This information was used to determine the average number and duration of contractions, as well as the average percentage of time spent by the rat contracting the abdomen during each 5 min period. The average number of contractions is represented per time unit (minutes).

### 2.4. Histological Damage

Colonic samples for histology were obtained from the same animals that underwent visceral sensitivity experiments (8–13 animals per group).

After sacrifice, 1 cm of colon was obtained at 8 cm depth (1 cm above the probe tip location), fixed in buffered 10% formalin, and embedded in paraffin. Sections of 5 μm were stained with hematoxylin–eosin (H/E) staining or toluidine blue and studied under a Zeiss Axioskop 2 microscope equipped with the image analysis software package AxioVision 4.6.

The analysis of colonic damage was made in ten random fields per section measured under a 40× objective. The histological damage of the colon was evaluated using a semi-quantitative scoring system adapted from Saccani et al. (2012) [19], as the sum of the following parameters: epithelial damage (0 to 3, normal to extensive), infiltration of inflammatory cells (0 to 4, absence to abundant infiltration), and goblet cell depletion (0 to 4, specific to general). Thus, maximum damage for colon was represented by 11 points.

### 2.5. Number of Mast Cells

The number of mast cells stained with toluidine blue [20] was counted under a 40× objective in ten fields per colonic sample all along the area between the epithelium and external muscular layer.

### 2.6. Immunohistochemistry

For immunohistochemistry, colonic samples were deparaffinized and rehydrated, and antigen unmasking was performed by boiling (98 °C) in 10 mmol/L citrate buffer for 30 min. Thereafter, sections were incubated for 10 min in 3% (vol/vol) in hydrogen peroxide to inhibit endogenous peroxidase activity and blocked with normal horse serum for 20 min to minimize nonspecific binding of the primary antibody. Sections were washed with phosphate buffered saline (PBS) with 0.05% Tween 20 (Calbiochem, Darmstadt, Germany) and then incubated overnight at 4 °C with the following primary antibodies: anti-macrophage-associated antigen CD163 (mouse anti-rat CD163, clone ED2, BioRAD; MCA342GA, 1/1000) to quantify the number of proinflammatory macrophages and anti-myeloperoxidase (MPO) to quantify neutrophils (rabbit anti-rat myeloperoxidase antibody carboxyterminal end; Abcam; ab65871; 1/1000).

The peroxidase kit ImmPRESS^®^ HRP Universal (Vector Laboratories Inc., Burkingame, Burlingame, CA, USA; horse anti-rabbit IgG plus polymer kit or horse anti-mouse IgG PLUS polymer kit) was used as secondary antibody. Samples were counterstained with hematoxylin and coverslips mounted with Eukitt mounting media (O. Kindler GmbH & Co., Freiburg, Germany). To determine the level of non-specific staining, negative controls were assessed.

### 2.7. Statistical Analysis

Statistical analyses were performed using Prism 8.0 (GraphPad Software Inc., La Jolla, CA, USA). Results are expressed as mean ± standard error of the mean (SEM). One-way (histological damage and immunohistochemical quantification) or two-way ANOVA (colonic sensitivity) followed by Bonferroni’s *post hoc* test were used for analyses. Differences were considered significant when *p* < 0.05.

## 3. Results

### 3.1. PHASE I: Effects of Acetic Acid and Choice of the Experimental Model

#### 3.1.1. Visceral Sensitivity

For evaluation of visceral sensitivity, the number of contractions per minute, the duration of abdominal contractions, and the percentage of time in contraction were analyzed (Figure 2).

As shown in Figure 2A, both control groups treated with SAL and evaluated 30 (SAL30) or 60 (SAL60) minutes after intracolonic administration showed an increase in the number of contractions as the intracolonic pressure increased, without statistically significant differences between them. Administration of 4% acetic acid 60 min before the test (ACE 4% 60 group) caused a marked decrease in the number of contractions at the highest pressures (60 and 80 mmHg), with statistically significant differences (*p* < 0.05 and *p* < 0.01, respectively) compared with SAL30 group, although differences to SAL60 group did not reach statistical significance. The reduction of the application time of acetic acid to 30 min maintaining the same dose (ACE 4% 30 group) resulted in a similar number of contractions to control groups (SAL30, SAL60), displaying statistically significant differences to the ACE 4% 60 group at 80 mmHg (*p* < 0.01). Finally, when both the dose of acetic acid and the time after exposure were reduced (ACE 0.6% 30), a slight non-significant reduction in the number of contractions was observed that was not as pronounced as that observed for the ACE 4% 60 group.

Regarding the duration of the abdominal contractions (Figure 2B), SAL30 control group showed a pressure-dependent increase in the duration of contractions except at 80 mmHg, where it slightly dropped. The SAL60 group exhibited a pressure-dependent increase during the experiment. Duration of contractions for ACE 4% 60 and SAL30 group were similar but ACE 4% 60 group showed statistically significant differences (*p* < 0.05) with SAL60 group at 80 mm Hg. For ACE 4% 30 group, values were similar to those observed in the SAL30 group, except for the lowest pressures (20 and 40 mmHg), where contractions were shorter. The ACE 0.6% 30 group showed a general decrease in duration of contractions with pressure and presented statistically significant differences for 60 and 80 mmHg (*p* < 0.05) with the ACE 4% 30 and SAL60 groups, respectively.

Finally, as can be seen in Figure 2C, the percentage of time in contraction for SAL30 (control) and SAL60 groups showed a pressure-dependent increase, with closely overlapping curves that did not show any statistically significant differences between them. Nevertheless, ACE 4% 60 group did not show this pressure-dependent increase observed for SAL30 and SAL60 but showed values remarkably lower than those obtained for these groups, with statistically significant differences to the SAL30 group at 60 and 80 mmHg (*p* < 0.05 and *p* < 0.01, respectively), and with SAL60 group for 80 mmHg (*p* < 0.01). In contrast, ACE 4% 30 group increased the percentage of time in contraction with pressure in a similar manner to controls, but with slightly lower values, and reaching statistically significant differences (*p* < 0.01) with the ACE 4% 60 group. In the case of the ACE 0.6% 30 group, the values obtained were more similar to those of the ACE 4% 60 group than to SAL groups or ACE 4% 30 group, although no statistically significant differences were found between ACE 0.6% 30 and the rest of the groups.

#### 3.1.2. Histological Damage

SAL30 and SAL60 control groups showed mild damage in colonic tissue, without statistically significant differences between them (Figure 3A). The results of ACE 4% 30 group showed a relatively small increase in tissue damage with statistically significant differences (*p* < 0.05) from the SAL60 group. Finally, the ACE 4% 60 and ACE 0.6% 30 groups showed slightly increased mean values of damage compared to controls (SAL30 and SAL60), although these differences did not reach statistical significance (Figure 3A). Mucosal damage was manifested by a loss of mucosal integrity, with damage to the epithelium and increased inflammatory infiltrates (Figure 3B).

#### 3.1.3. Number of Mast Cells

The number of mast cells in the SAL30 group was the lowest. Compared with SAL 30, SAL60 showed an increase in the number of mast cells but the difference did not reach statistical significance. The groups treated with acetic acid at the highest concentration (4%) showed an increase in the number of mast cells, but the difference was statistically significant only for ACE 4% 30 compared with SAL30 (*p* < 0.01). When the concentration of acetic acid was reduced to 0.6%, the number of mast cells decreased accordingly, without any statistically significant difference to SAL 30 (Figure 4).

#### 3.1.4. Immunohistochemistry

The number of macrophages labeled with anti-CD163 antibody (Figure 5) was similar in both control groups (SAL30 and SAL60). Intracolonic administration of acetic acid (4%) increased the number of macrophages regardless of the time elapsed after its administration, but the differences between the control groups did not reach statistical significance. In the animals treated with acetic acid at the low concentration, there was a remarkable reduction in the number of macrophages, with statistically significant differences compared with the two treatments using 4% acetic acid (*p* < 0.05).

The analysis of the number of neutrophils labeled with anti-myeloperoxidase antibody (Figure 6) did not reveal any differences among the different treatments.

### 3.2. PHASE II: Evaluation of the Effect of a Commercial Probiotic

Considering all the data obtained in phase I, we proceeded to choose the model that would be used in the second phase of the study. The result for control groups were similar, regardless of the time of administration of SAL. Treatment with acetic acid produced clear functional modifications versus control groups in ACE 4% 60 and ACE 0.6% 30 groups.

Of these, ACE 0.6% 30 protocol was selected for phase II experiments because, in addition to significant functional changes, it produced significant changes in immunocytes (macrophages). At the same time, due to the short contact time of acetic acid, it would allow the earliest changes to be evaluated.

#### 3.2.1. Visceral Sensitivity

In phase II, two experimental groups (SAL + ACT; ACE + ACT) were added to the two groups selected from the first phase (SAL30 = SAL; ACE 0.6% 30 = ACE).

The number of abdominal contractions per minute (Figure 7A) in the SAL group showed a pressure-dependent increase, and previous administration of Actimel^®^ (SAL + ACT) produced a very similar curve, with no statistically significant differences versus the control group. As mentioned above, there was a slight reduction in the response observed in the ACE group, and this was prevented by prior administration of Actimel^®^ (ACE + ACT group), without statistically significant differences between ACE + ACT and either SAL or SAL + ACT groups.

The duration of the abdominal contractions (Figure 7B) in SAL and SAL + ACT groups showed a pressure-dependent increase, with values that were not significantly different between groups. Again, the ACE group exhibited a shorter duration of contractions, and showed statistically significant differences (*p* < 0.05) with the SAL group. In the case of the ACE + ACT group, the effects of ACE were not apparent, with values similar to SAL and SAL + ACT groups.

The percentage of time in contraction (Figure 7C) for SAL group and the SAL + ACT gradually increased in response to intracolonic pressure and returned practically to 0 when mechanical stimulation was withdrawn. The ACE group showed a notable decrease of this parameter, displaying statistically significant differences (*p* < 0.05) to the SAL group at 60 and 80 mm Hg. Again, the ACE + ACT group showed a behavior more similar to the SAL and SAL + ACT groups than to the pathological control ACE, showing statistically significant differences (*p* < 0.05) to it at 80 mmHg.

#### 3.2.2. Histological Damage

The administration of Actimel^®^ alone (SAL + ACT) tended to slightly decrease tissue damage, without statistically significant differences to the SAL group. As previously described, the ACE group had slightly higher rates of colonic damage than the SAL group without reaching statistically significant differences, but the difference to the SAL + ACT group was statistically significant (*p* < 0.05). Pretreatment with Actimel^®^ slightly reduced the damage in the ACE + ACT group without statistically significant differences to the other three experimental groups (Figure 8).

#### 3.2.3. Number of Mast Cells

No significant differences were found between the experimental groups, but there was a slight increase in the number of mast cells in the SAL + ACT group (Figure 9).

#### 3.2.4. Immunohistochemistry

The number of macrophages (Figure 10) was reduced in the samples treated with acetic acid while pretreatment with Actimel^®^ increased the number of macrophages. It should be noted that there was also an increase in the number of macrophages in the SAL + ACT samples.

There were not remarkable changes in the number of neutrophils among the experimental groups evaluated in phase II (Figure 11).

## 4. Discussion

The present study shows that intracolonic administration of the irritant acetic acid may cause an early decrease in the number of abdominal contractions in response to mechanical stimulation applied to the colon. This is associated with changes in the number of immunocytes, particularly macrophages. Additionally, it also shows that pretreatment with a commercial preparation with probiotics (Actimel^®^) may prevent the effects caused by the irritant on both visceral sensitivity and this inflammatory cell population.

It has been reported that intracolonic administration of acetic acid at low concentrations (<1%) causes sensitization of afferent neurons to mechanical stimulation with absence of inflammation of the colonic mucosa (IBS model). In contrast, higher concentrations of this irritant may provoke inflammation of the tissue and an increase in myeloperoxidase (MPO, an enzyme present in neutrophils) activity, as well as in visceral hypersensitivity once the inflammatory response ceases (IBD model). In fact, in animal models, elevated concentrations of acetic acid are usually used [21,22,23,24].

In the study carried out by Dai et al. (2012) [25], 7 days after intracolonic administration of 4% acetic acid, an increased response to rectal distention, as well as a greater epithelial permeability, was reported. Li et al. (2019) [26] also observed a significant increase in sensitivity to intracolonic mechanical stimulation and in the number of mast cells, 7 days after administration. In the experiment carried out by Wu et al. (2017) [27], 8 days after administration of 4% acetic acid, a significant increase in the abdominal response to colonic stimulation was detected accompanied by a considerable increase in MPO activity and the recruitment of neutrophils and lymphocytes on the second day after administration. In contrast with these studies, in the present research we have observed a decrease in the abdominal response to intracolonic mechanical stimulation, especially in ACE 0.6% 30 and ACE 4% 60 groups. This disparity could be due to methodological differences; in the aforementioned studies, the tests employed to evaluate visceral sensitivity were performed one week after induction of colitis, once the inflammation process had resolved. In our case, the inflammation/irritation process was evaluated at a very early stage of its development (30 to 60 min after the administration of the irritant), which could explain the differences in visceral responses. The reduction observed is probably transitory and could disappear at later times, even changing into the typical hypersensitivity that other studies and models have shown. This is something to be corroborated with specific studies in our model.

Another important difference resides in the pattern of mechanical stimulation during the test. In the above-mentioned studies, animals were subjected to phasic stimulation, with pressure intervals applied for only 20–30 s with 2, 4, and 5 min breaks in between. In our experiment, we have used a tonic stimulation protocol: each pressure stimulus was applied for 5 min, with no rest time between different pressures. Animals may respond differently to prolonged (tonic) intracolonic stimulations compared to shorter (phasic) periods of stimulation [18], perhaps due to habituation to the stimulus or the reduction of pain sensitivity as a consequence of the release of endogenous opioids [28,29,30,31] or other inhibitory neurotransmitters or neuromodulators (endocannabinoids, somatostatin [32,33,34]) in response to the irritant. Furthermore, it has been observed that colonic inflammation causes the activation of the vagus nerve, which produces an anti-inflammatory reflex, decreasing the release of proinflammatory cytokines [35].

Despite having also received a high dose of acetic acid, ACE 4% 30 did not exhibit this decreased response to mechanical stimulation that ACE 4% 60 group did. It is possible that pain inhibitory mechanisms may take longer to activate, which would explain its absence 30 min after administration of acetic acid. It is necessary to study the effects of this high dose at different times to better determine the dynamics of the changes that occur in these early times (minutes) after the administration.

Regarding low concentrations of acetic acid in the literature, the administration of 1.5 mL of 0.6% acetic acid caused an increase in the number of abdominal contractions in response to tonic mechanical intracolonic stimulation one hour after instillation [36]. Langlois et al. (1994) [21] observed that the administration of 1 mL of 0.6% acetic acid decreased the pressure needed to cause a cardiovascular response and an increase in the response magnitude 45 min after application. Plourde et al. (1997) [37] detected a significant increase in the number of abdominal contractions in rats subjected to intracolonic stimulation one hour after intracolonic administration of 1.5 mL of 0.6% acetic acid. The administration of 0.2 mL of 0.5% acetic acid [38] increased the abdominal response to intracolonic stimulation (0–80 mmHg) 6 weeks after irritant application.

In our study, intracolonic administration of 0.7 mL of 0.6% acetic acid produced a decrease in the response to intracolonic mechanical stimulation 30 min after administration. The disparity with the aforementioned experiments could be explained taking into account the existing methodological differences. In the cited studies, the intracolonic mechanical stimulation pattern is again different to the one we have used. Russell et al. (2018) [36] maintained the stimulation for 10 min with a 10 min rest period between pressures; Langlois et al. (1994) [21] maintained it for 30 s with 5 min breaks; Plourde et al. (1997) [37] maintained it for 10 min, without alternating pressures; and Zhu et al. (2015) [38] maintained the stimulus for 20 s with 2 min rests between pressures, in addition to performing the experiments 6 weeks after the application of the irritant. It is undeniable that the design of the experiment greatly influences the observed functional results.

Whatever the case may be, the mechanisms mentioned above for acetic acid at 4% could be participating also with low doses of acetic acid. To verify the activation of such mechanisms, it would be interesting to evaluate, for example, the effects of the administration of cannabinoid or opioid antagonists that could block them.

Regarding the histological results observed in the colonic samples, acetic acid caused a slight increase in the tissue damage values, bigger for higher concentrations (ACE 4% > ACE 0.6%) and for shorter treatment duration (ACE 4% 30 > ACE 4% 60). The results seem to suggest that irritation of the mucosa, although important, is still at very early stages. In any case, the probable rupture of the barrier function of the epithelium could be generating a stimulating effect on the sensory afferent endings. In the pathogenesis of IBS, low-grade inflammation is considered to be an important issue, whereas in IBD there is a chronic inflammation. To verify if inflammation was present, we analyzed the populations of immune cells in the colon after visceral pain experiments.

The number of mast cells is usually increased in IBS and IBD patients, and these cells participate in the sensitization of afferent fibers related to the presence of visceral pain in these patients [39,40,41]. Several authors, such as Li et al. (2019) [26], observed a significant increase in sensitivity to intracolonic mechanical stimulation with a correlative increase in the number of mast cells 7 days after administration. In the experiments performed by Wu et al. (2017) [27], administration of 4% acetic acid caused a significant increase in the abdominal response to colonic stimulation after 8 days, which was correlated with an increase in the number of mast cells. Karaca et al. (2010) [42] also found an increase in the number of mast cells following intracolonic administration of 4% acetic acid one month after colitis induction. It should be noted that in all these cases, the analyses were performed at much later times than in our experiments.

In our study, despite acetic acid at high concentrations (ACE 4% 30) increased the number of colonic mast cells, the visceral pain responses were similar to those obtained in control animals intracolonically treated with saline, may be due to an early activation of the aforementioned inhibitory mechanisms. In contrast, the low concentration of acetic acid (0.6%) did not cause an increase in the number of mast cells, but reduced visceral pain responses to intracolonic mechanical stimulation, suggesting the involvement of a mast cell independent mechanism in the inhibition of visceral pain responses, which needs further investigation.

A tendency to increase the number of proinflammatory M2 macrophages was observed in the samples treated with the highest doses of the irritant (4%). In addition, this treatment was the one with the highest number of mast cells. The products secreted by mast cells stimulate macrophage migration to areas of inflammation [39,41], which could explain this increase. In contrast, there was a reduction in the number of activated macrophages in samples from animals treated with 0.6% acetic acid for 30 min that occurred together with a reduction in the number of mast cells.

No changes were observed in the number of neutrophils between the different treatments. This may again be due to the relatively short application times of the irritant, maybe too short for these cells to activate. Other authors [22,27,43,44] detected an increase in MPO expression after performing intracolonic treatments with 4% acetic acid. However, measurements for MPO detection were made several hours or even days after treatment, in contrast with our early protocol. This would indicate that more time could be needed to detect changes in the population of these cells.

In recent years, probiotics have emerged as a possible effective tool for the treatment of diseases associated with dysbiosis of the human microbiota and have been the subject of numerous meta-analysis [45,46]. Despite its broad use by people of all ages and conditions, very few studies have evaluated the effects of Actimel^®^ (the commercial probiotic used here). In the study by Hunsche et al. (2018) [47], supplementation with Actimel^®^ in old mice caused improvements in behavioral aspects (balance, muscular vigor, motor coordination, anxiety-type behaviors, among others) and in immunological aspects, both in the short (1 week) and long (4 weeks) term, obtaining values similar to adult controls. Importantly, in the context of our research, positive effects of the administration of probiotics have been observed in the gastrointestinal tract, such as the reduction of visceral hypersensitivity and abdominal pain. A widely studied probiotic is the *Bifidobacterium longum spp. infantis* 35624 strain [48]. In fact, it is the strain present in the only approved probiotic drug, to date, for the treatment of IBS (Alflorex^®^). However, the results are conflicting, and the American Gastroenterology Association (AGA) considers that there is a “very low quality of evidence” to recommend its use [49].

Numerous studies have observed antinociceptive and anti-inflammatory effects after probiotic administration in animal models of colonic inflammation [16,50]. In the study by Dai et al. (2011) [25], the administration of a set of eight strains (now termed “De Simone formulation”, previously known as VSL#3) in a colitis model developed using acetic acid significantly attenuated visceral hypersensitivity and colonic epithelial permeability. In a study in rats subjected to stress by movement restriction (visceral hypersensitivity IBS-model associated with stress), the probiotic *Bifidobacterium lactis* (present in the commercial preparation used in this work) decreased the visceral response (number of abdominal contractions) and the intestinal permeability [50]. A reduction in visceral pain has also been observed in clinical trials in IBS patients when probiotics such as the *Lactiplantibacillus plantarum* 299 v strain were used [51]. These studies confirm that probiotics can be effective in the treatment of gastrointestinal diseases that present with abdominal pain.

Taking all this into account, in the second phase of the study, the effects of a pretreatment with Actimel^®^ on colonic irritation/low inflammation were evaluated. The selected group for this part of the study was ACE 0.6% 30 (ACE) for practical reasons (less time required) and, importantly, for displaying (thus far unreported) an inhibitory response at such early time after colonic irritation. It was also the treatment in which the changes in the number of activated macrophages were most significant. The treatment with Actimel^®^ prevented the effects caused by acetic acid in visceral sensitivity, producing values similar to those shown for the control group (SAL), which received no irritant. Regarding the damage observed during the tissue histological evaluation, Actimel^®^ also tended to decrease the levels of damage for both the control group and for the ACE group.

When pretreatment with Actimel^®^ was performed, no significant differences in the number of mast cells were detected. Moreover, in the analysis of visceral pain, pretreatment with Actimel^®^ in animals treated with 0.6% acetic acid (ACE) caused a recovery of visceral pain responses to values similar to those of the SAL group. Taking into account that the amount of mast cells in the ACE and ACE + ACT samples is similar, it is possible that the modifications observed in visceral sensitivity are not so much due to the sensitization of the afferent fibers by the action of these cells, but rather to modifications in the inhibitory mechanisms mentioned above. Perhaps the effect of pretreatment with Actimel^®^ reduces the triggering of these inhibitory mechanisms, recovering values similar to those of the control groups.

In the immunohistochemistry study, no changes in the number of neutrophils were found with Actimel^®^ pretreatment. As mentioned above, the times used in these experiments may not be sufficient to alter this population of immunocytes.

In contrast, there was an increase in the number of macrophages in the animals treated with Actimel^®^, both in SAL + ACT and ACE + ACT. In a study performed by Hunsche et al. [47], it was observed that supplementing the diet of old mice with fermented milk (*Lacticaseibacillus casei* DN-114001) for 1 week improved the functions of macrophages by increasing their migration and phagocytosis capacity. Another study, in which the diet of young mice was supplemented with the same ferment for 98 days, showed that after 14 days there was an increase in the phagocytic capacity of peritoneal macrophages, showing the activation of cells far from the site where the stimulation occurred [52]. Studies in healthy patients showed that the application of a probiotic (*Bifidobacterium lactis* Bi-07) in four periods of 3 weeks separated by periods of 4 weeks led to an improvement in the phagocytic activity of monocytes and granulocytes [53]. All this seems to indicate that treatment with probiotics has a direct effect on these immune functions, stimulating their activity.

Thus, the commercial product used here was able to counteract the functional (inhibition of abdominal contractions) and histological changes produced early after intracolonic administration of acetic acid (and it would be necessary to evaluate its effects at later stages of acetic acid-induced colonic irritation). This could be due to the probiotic itself, but other components present in the commercial preparation administered, such as vitamin D, could have influenced the results. In fact, deficiencies in vitamin D levels have been observed in IBD patients, and there is evidence that the administration of vitamin D could be beneficial against colitis, reducing inflammation, inhibiting the cell proliferation and apoptosis, in addition to regulating the gastrointestinal microbiota [54]. However, the animals used in the study were not deficient in vitamin D. Furthermore, since doses of vitamin D of 12.5 mg/kg were not toxic to rats [55,56], the amount of vitamin D daily administered to our animals with Actimel^®^ (calculated as 0.06 mg/kg) probably did not cause significant changes, especially considering the relatively short duration of pretreatment. In any case, the contribution of the non-probiotic components of Actimel^®^ (vitamin B9, iron, zinc) to the effects observed in this research requires specific investigation.

Finally, our study has allowed us to obtain a first approximation to the effects of Actimel^®^ on visceral pain and immunocytes in young adult males. Nevertheless, in the last years, it has been demonstrated that males and females do not behave in the same way in visceral pain experiments [18,20], and also different responses may be found in animals of different ages. Since abdominal pain is a highly prevalent condition irrespective of sex and age, and Actimel ^®^ is also a popular commercial product, further research investigating the influence of these factors (sex and age) is warranted.

## 5. Conclusions

In the present study, we have analyzed the effects produced by the intracolonic administration of acetic acid which, contrary to initial expectations, produced a decrease in the abdominal response to mechanical colonic stimulation and a slight increase in the damage to the colonic mucosa shortly after administration (30/60 min). These results could be explained through a possible early activation of inhibitory mechanisms, to our knowledge, not previously described in visceral pain models. However, to verify this possibility, future investigations, using antagonists of the receptors most likely involved in it, are necessary.

Additionally, pretreatment with probiotics prevented the effects caused by acetic acid, causing a return to values such as those of the control group. Therefore, our results suggest that probiotics may have the ability to avoid not only the excitatory effects reported in the literature to occur at later stages after mucosal irritation, but also the early inhibitory effect of acetic acid colonic irritation on visceral sensitivity occurring shortly after administration, as described in this study.

Further research is needed to determine the influence of factors such as age and sex, as well as the mechanisms involved in the effects observed here.

## Figures and Tables

**Figure 1 ijerph-19-06485-f001:**
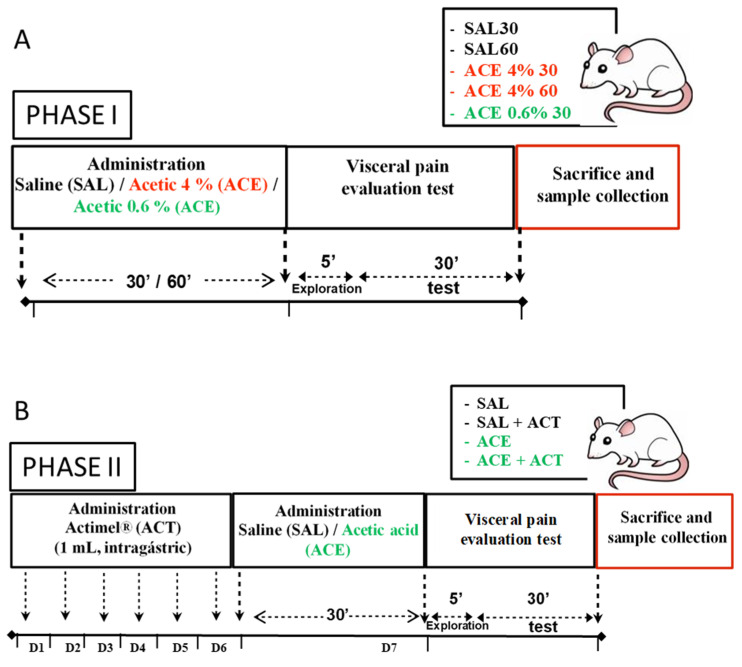
Experimental protocol. Young male Wistar rats were used. The study was divided into two phases. In both, functional (visceral sensitivity) and structural (colonic wall histology and immunohistochemistry) effects were evaluated. (**A**) Phase I, selection of the experimental model. For this, animals were administered intracolonic saline (control, SAL) or acetic acid (ACE) at two different concentrations: 0.6% (ACE 0.6%, IBS model, green); 4% (ACE 4%, IBD model, red). Visceral pain was evaluated 30 (SAL30/ACE30) or 60 (SAL60/ACE60) minutes after SAL/ACE administration (early stages of the models). (**B**) Phase II, evaluation of the effects of the prior administration of probiotics on the selected model (ACE 0.6% 30). For this, Actimel^®^ (ACT) was intragastrically administered at 1 mL/day for the 7 days prior to the performance of the visceral sensitivity test. Four experimental groups were evaluated in phase II, according to the substance intracolonically administered and whether they received ACT or not: SAL, SAL + ACT, ACE, and ACE + ACT.

**Figure 2 ijerph-19-06485-f002:**
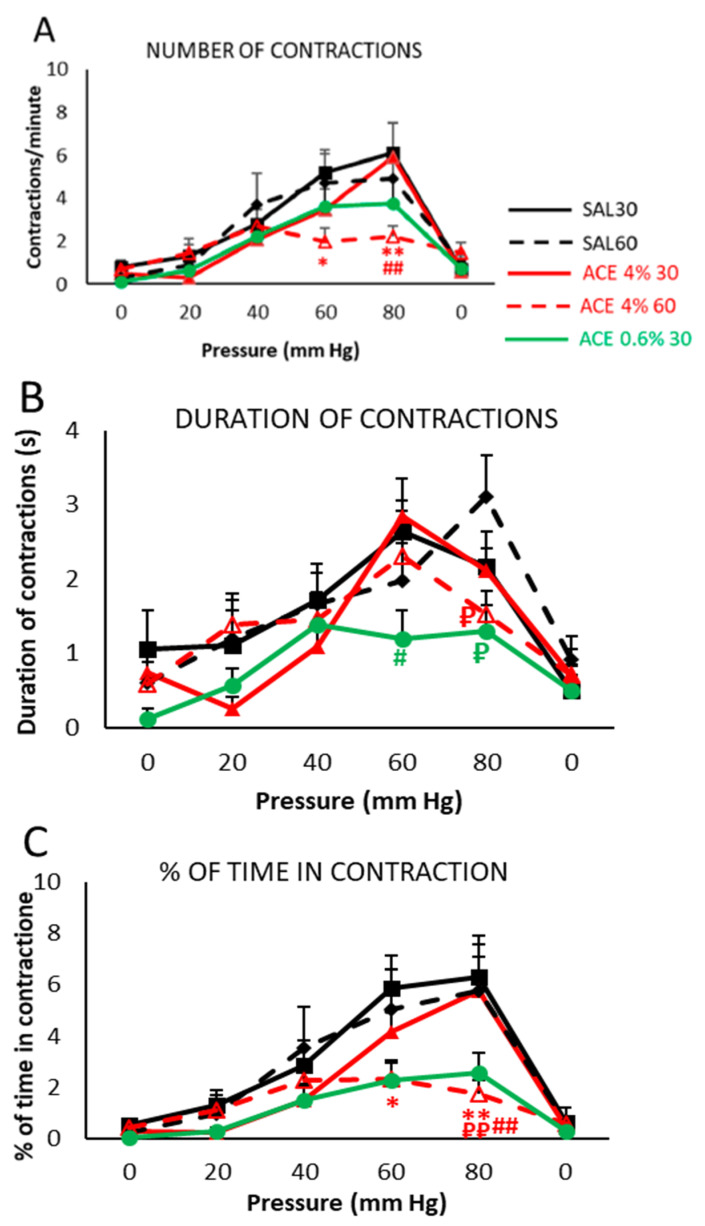
Effect of intracolonic administration of acetic acid on visceral sensitivity, phase I (model selection). Animals were subjected to tonic mechanical intracolonic stimulation. Pressure was increased from 0 to 80 mm Hg, in steps of 20 mm Hg every 5 min, to finally return to 0 mm Hg again; for each pressure value, a single stimulus was applied and maintained for 5 min. Number of contractions per minute (**A**), duration of contractions (**B**), and percentage of time in contraction (**C**) were measured. Animals (*n* = 8–13/group) were intracolonically administered with saline (control, SAL) or acetic acid (ACE) at two different concentrations: 0.6% or 4%. The visceral sensitivity test was performed at two different time points after administration: 30 (SAL30, black line/ACE 0.6% 30, green line/ACE 4% 30, red line) or 60 min (SAL60, black dotted line /ACE 4% 60, red dotted line). Data represent the mean ± SEM. * *p* < 0.05 vs. SAL30; ** *p* < 0.01 vs. SAL30; ₽ *p* < 0.05 vs. SAL60; ₽₽ *p* < 0.01 vs. SAL60; # *p* < 0.05 vs. ACE 4% 30, ## *p* < 0.01 vs. ACE 4% 30. Two-way ANOVA followed by Bonferroni *post hoc* test (*n* = 8–13).

**Figure 3 ijerph-19-06485-f003:**
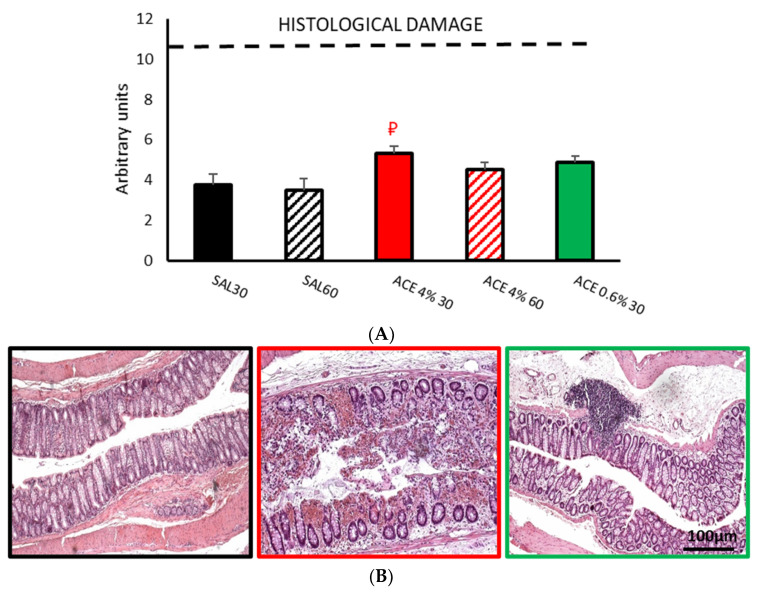
Effect of intracolonic administration of acetic acid on histological damage of colonic wall, phase I (model selection). Histological damage (**A**) was measured in ten random fields per section. Maximum damage was represented by 11 points (dotted line). Experimental groups: SAL30, black bar; SAL60, black striped bar. ACE 4% 30, red bar; ACE 4% 60, red striped bar; ACE 0.6% 30, green bar. Data represent mean ± SEM. ₽ *p* < 0.05 vs. SAL60. One-way ANOVA followed by Bonferroni *post hoc* test. (*n* = 8–13). (**B**) Photomicrographs of colonic samples stained with H/E after intracolonic administration of SAL30 (black frame), ACE 4% 30 (red frame), or ACE 0.6% 30 (green frame). Scale bar = 100 μm.

**Figure 4 ijerph-19-06485-f004:**
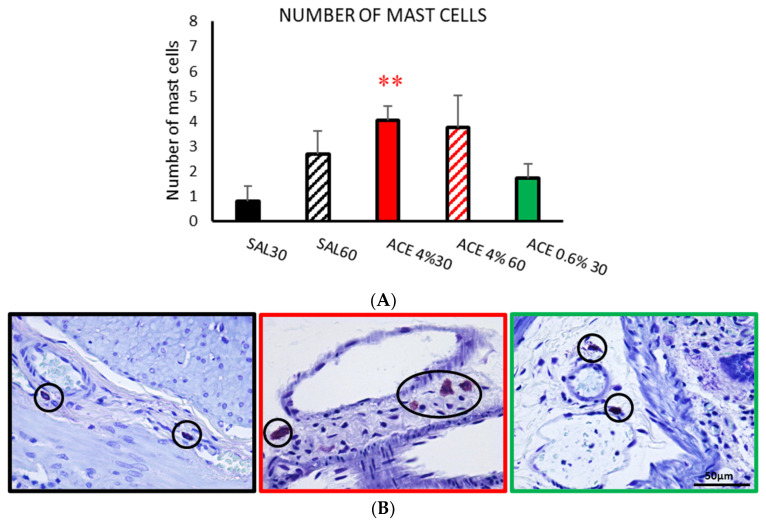
Effect of intracolonic administration of acetic acid on the number of mast cells, phase I (model selection). Colon sections of were stained with toluidine blue and studied under a Zeiss Axioskop 2 microscope. The number of mast cells (**A**) was counted in ten random fields per section across mucosa and submucosa. Experimental groups: SAL30, black bar; SAL60, black striped bar; ACE 4% 30, red bar; ACE 4% 60, red striped bar; ACE 0.6% 30, green bar. Data represent mean ± SEM. ** *p* < 0.01 vs. SAL30. One-way ANOVA followed by Bonferroni *post hoc* test (*n* = 8–13). (**B**) Photomicrographs of mast cells stained with toluidine blue (circles) after intracolonic administration of SAL30 (black frame), ACE 4% 30 (red frame), or ACE 0.6% 30 (green frame). Scale bar = 50 μm.

**Figure 5 ijerph-19-06485-f005:**
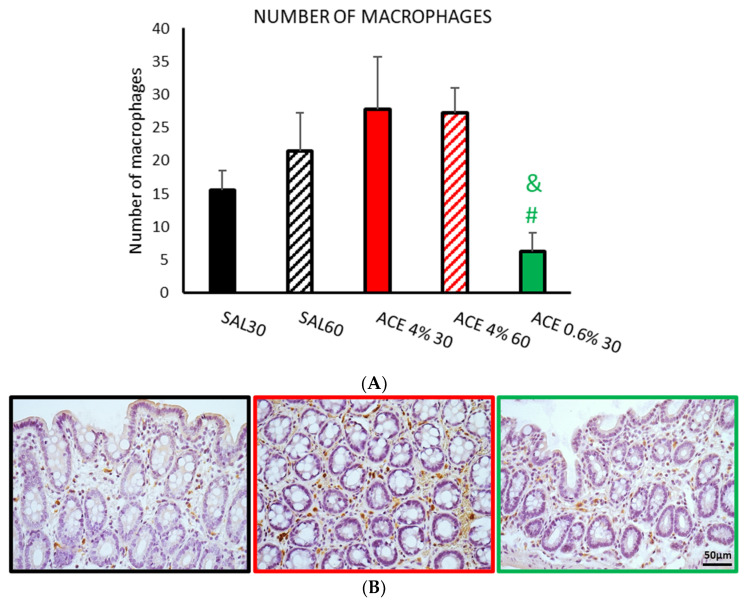
Effect of intracolonic administration of acetic acid on the number of macrophages, phase I (model selection). The number of macrophages (**A**) was assessed by immunohistochemistry in ten random fields per section. Experimental groups: SAL30, black bar; SAL60, black striped bar; ACE 4% 30, red bar; ACE 4% 60, red striped bar; ACE 0.6% 30, green bar. Data represent mean ± SEM. # *p* < 0.05 vs. ACE 4% 30; & *p* < 0.05 vs. ACE 4% 60. One-way ANOVA followed by Bonferroni *post hoc* test (*n* = 8–13). (**B**) Photomicrographs of CD163 immunopositive cells (pro-inflammatory macrophages) in colonic mucosa after intracolonic administration of SAL30 (black frame), ACE 4% 30 (red frame), or ACE 0.6% 30 (green frame). Scale bar = 50 μm.

**Figure 6 ijerph-19-06485-f006:**
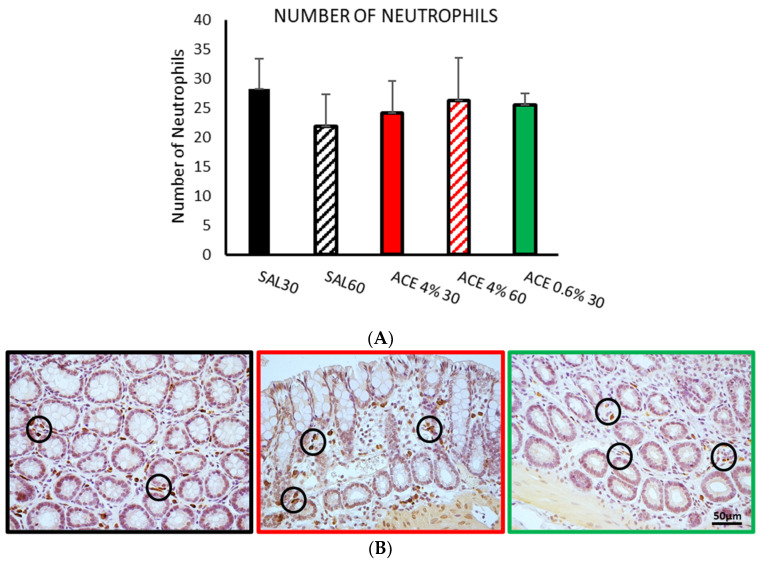
Effect of intracolonic administration of acetic acid on the number of neutrophils, phase I (model selection). The number of neutrophils (**A**) was assessed by immunohistochemistry and studied in ten random fields per section. Experimental groups: SAL30, black bar; SAL60, black striped bar; ACE 4% 30, red bar; ACE 4% 60, red striped bar; ACE 0.6% 30, green bar. Data represent mean ± SEM. One-way ANOVA followed by Bonferroni *post hoc* test (*n* = 8–13). (**B**) Photomicrographs of myeloperoxidase immunopositive cells (neutrophils, see circles) in colonic mucosa after intracolonic administration of SAL30 (black frame), ACE 4% 30 (red frame), or ACE 0.6% 30 (green frame). Scale bar = 50 μm.

**Figure 7 ijerph-19-06485-f007:**
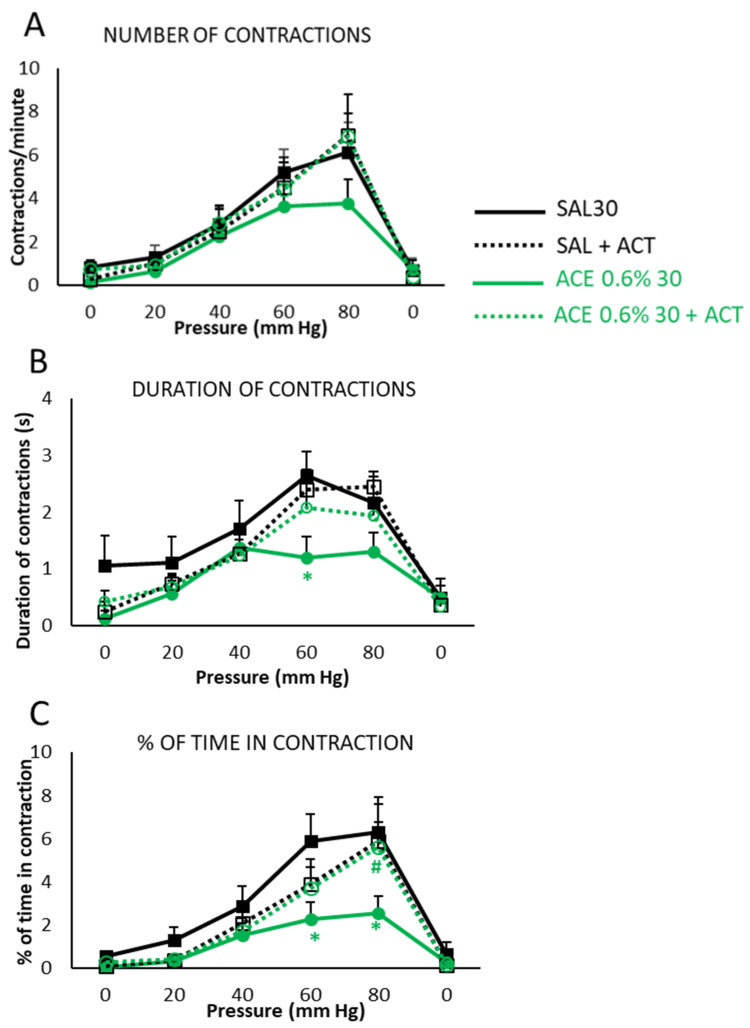
Effects of pretreatment with probiotics on visceral sensitivity, phase II. The groups selected in phase I (SAL30 and ACE 0.6% 30, now termed SAL, black line and ACE, green line) were compared with the corresponding two groups that were pretreated with 1 mL/day of Actimel^®^ (ACT) 7 days before performance of the visceral sensitivity test (SAL + ACT black dotted line, ACE + ACT green dotted line). Animals were subjected to tonic mechanical intracolonic stimulation. Pressure was increased from 0 to 80 mm Hg, in steps of 20 mm Hg every 5 min, to finally return to 0 mm Hg again; for each pressure value, a single stimulus was applied and maintained for 5 min. Number of contractions per minute (**A**), duration of contractions (**B**), and percentage of time in contraction (**C**) were measured. Data represent the mean ± SEM. * *p* < 0.05 vs. SAL; # *p* < 0.05 vs. ACE. Two-way ANOVA followed by Bonferroni *post hoc* test (*n* = 8–10).

**Figure 8 ijerph-19-06485-f008:**
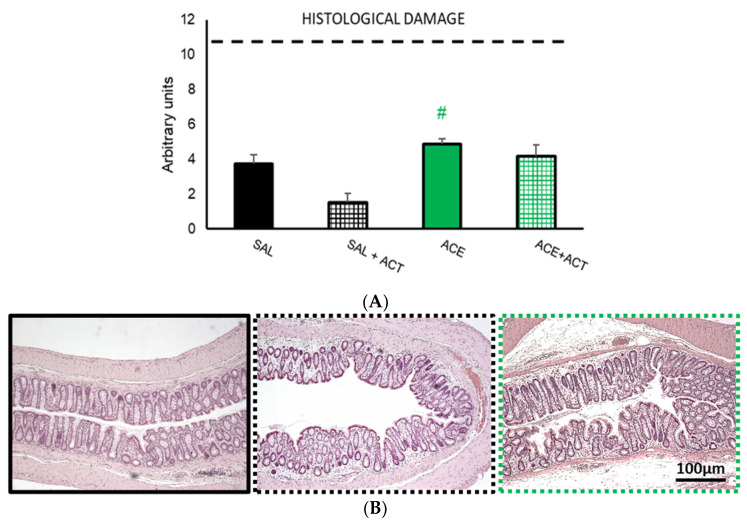
Effects of pretreatment with probiotics on colonic damage, phase II. The groups selected in phase I (SAL30 and ACE 0.6% 30, now termed SAL, black bar and ACE, green bar) were compared with the corresponding two groups that were pretreated with 1 mL/day of Actimel^®^ (ACT) (SAL + ACT black gridded bar, ACE + ACT green gridded bar). Histological damage (**A**) was measured in ten random fields per section stained with HE. Maximum damage was represented by 11 points (dotted line). Data represent mean ± SEM. # *p* < 0.05 vs. SAL + ACT. One-way ANOVA followed by Bonferroni *post hoc* test (*n* = 8–13). (**B**) Photomicrographs of colonic samples stained with H/E after intracolonic administration of SAL (black frame) or in animals pretreated with Actimel^®^ after intracolonic administration of saline (black dotted frame) or 0.6% acetic acid (green dotted frame). Scale bar = 100 μm.

**Figure 9 ijerph-19-06485-f009:**
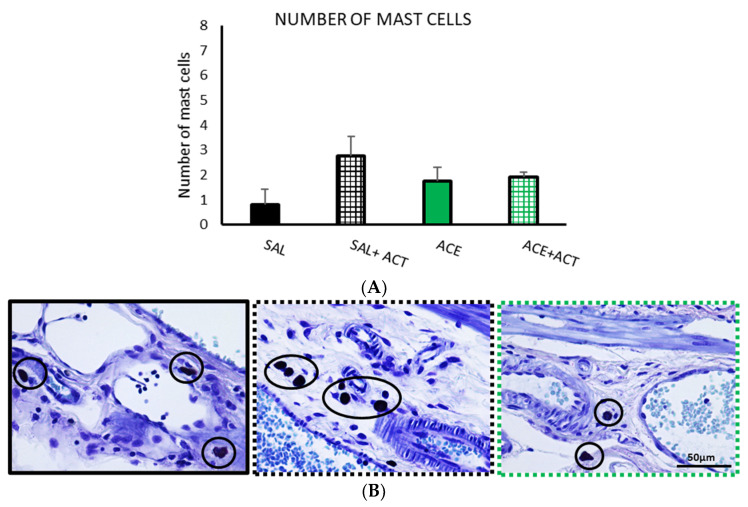
Effects of pretreatment with probiotics on the number of mast cells, phase II. The groups selected in phase I (SAL30 and ACE 0.6% 30, now termed SAL, black bar and ACE, green bar) were compared with the corresponding two groups that were pretreated with 1 mL/day of Actimel^®^ (ACT) (SAL + ACT black gridded bar, ACE + ACT green gridded bar). (**A**) The number of mast cells was counted in ten random fields per section across mucosa and submucosa. Data represent mean ± SEM. One-way ANOVA followed by Bonferroni *post hoc* test (*n* = 8–13). (**B**) Photomicrographs of colonic samples stained with toluidine blue for mast cell detection (circles) after intracolonic administration of SAL (black frame) or in animals pretreated with Actimel^®^ after intracolonic administration of saline (SAL + ACT; black dotted frame) or 0.6% acetic acid (ACE + ACT; green dotted frame). Scale bar = 50 μm.

**Figure 10 ijerph-19-06485-f010:**
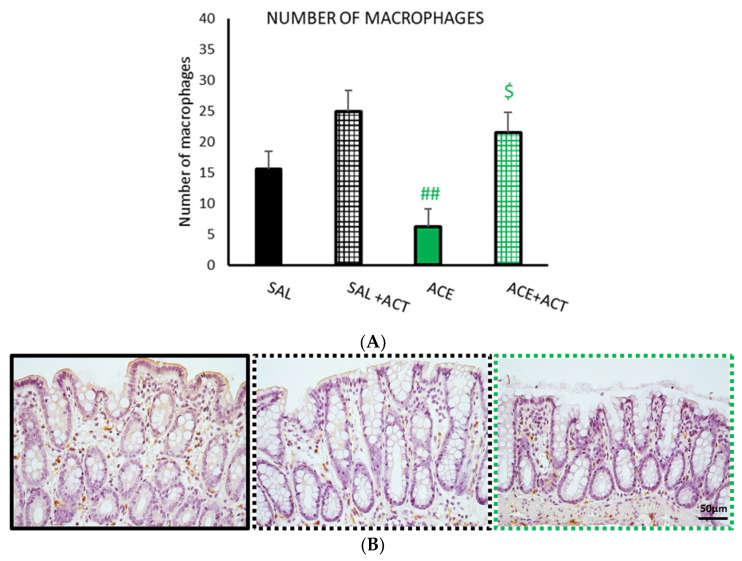
Effects of pretreatment with probiotics on the number of macrophages, phase II. The groups selected in phase I (SAL30 and ACE 0.6% 30, now termed SAL, black bar and ACE, green bar) were compared with the corresponding two groups that were pretreated with 1 mL/day of Actimel^®^ (ACT), the 7 days before performance of the visceral sensitivity test (SAL + ACT black gridded bar, ACE + ACT green gridded bar). The number of macrophages (**A**) was assessed using immunohistochemical methods in ten random fields per section. Data represent mean ± SEM. ## *p* < 0.01 vs. SAL+ ACT; $ *p* < 0.05 vs. ACE + ACT. One-way ANOVA followed by Bonferroni *post hoc* test. (**B**) Photomicrographs of CD163 immunopositive cells (macrophages) in colonic mucosa after intracolonic administration of SAL (black frame) or in animals pretreated with Actimel^®^ after intracolonic administration of saline (SAL + ACT; black dotted frame) or 0.6% acetic acid (ACE + ACT; green dotted frame). Scale bar = 50 μm.

**Figure 11 ijerph-19-06485-f011:**
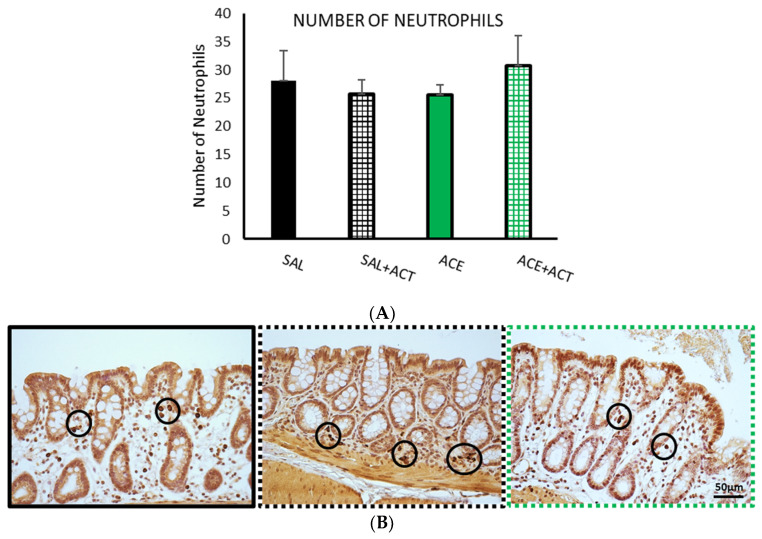
Effects of pretreatment with probiotics on the number of neutrophils, phase II. The groups selected in phase I (SAL30 and ACE 0.6% 30, now termed SAL, black bar and ACE, green bar) were compared with the corresponding two groups that were pretreated with 1 mL/day of Actimel^®^ (ACT), 7 days before performance of the visceral sensitivity test (SAL + ACT black gridded bar, ACE + ACT green gridded bar). The number of neutrophils (**A**) was assessed using immunohistochemical methods in ten random fields per section. Data represent mean ± SEM. No statically significant differences were found. One-way ANOVA followed by Bonferroni *post hoc* test (*n* = 8–13). (**B**) Photomicrographs of myeloperoxidase immunopositive cells (neutrophils, see circles) in colonic mucosa after intracolonic administration of SAL (black frame) or in animals pretreated with Actimel^®^ after intracolonic administration of saline (SAL + ACT; black dotted frame) or 0.6% acetic acid (ACE + ACT) green dotted frame). Scale bar = 50 μm.

## Data Availability

Data will be available upon reasonable request to the corresponding authors (J.A.U., R.A.).

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
