# Peer review of "Effects of Commercial Probiotics on Colonic Sensitivity after Acute Mucosal Irritation"

_ijerph, 2022, doi:10.3390/ijerph19116485_

Round 1
Reviewer 1 Report
Brief summary
This is an interesting original experimental article, carried out in a rat animal model. In this work, the early consequences of intracolonic administration of the irritant acetic acid (ACE) on visceral sensitivity, colonic wall histology and some cells involved in inmune response have been analyzed. The intensity of these changes is variable and depends on the concentration of ACE used (4% or 0.6%) and the application time (30 or 60 minutes). Furthermore, the effects of pretreatment with a commercial probiotic (Actimel®) have been studied. It is shown that the application of ACE decreases the nociceptive response to intracolonic mechanical stimulation, with a slight increase in histological damage to the colonic mucosa. A decrease in the number of macrophages is also observed (only at the lowest dose of acetic acid and at the shortest time) without significantly altering the number of neutrophils or mast cells. It is interesting to emphasize that the early decrease in visceral sensitivity induced by ACE has not been previously described in the scientific literature.
In addition, the previous treatment with the commercial probiotic Actimel® causes a recovery of responses to visceral pain induded by ACE, to values similar to controls, and acts on immune functions by inducing an increase in the number of macrophages.
The experiments have been carried out with an adequate methodology and in accordance with the ethical standards of animal research.
General comments
The early decrease in visceral sensitivity induced by acetic acid, not previously described in the scientific literature, is explained through the possible early activation of inhibitory nociceptive mechanisms (release of endogenous opioids, activation of TRPV1 receptors ...) although this is a hypothesis, which is not confirmed experimentally in this article. This fact must be clearly reflected in the abstract and conclusions. As the authors suggest, to verify the activation of the aforementioned inhibitory mechanism, it would be necessary to evaluate various antagonists of the receptors involved in it (opioids, cannabinoids...) which will probably be the target of future research.
A previous study has been carried out to assess the effects of intracolonic administration of ACE at two different concentrations (0.6 or 4%), and at two different times after administration (30 or 60 minutes). Based on the results obtained, an experimental protocol with the lowest concentration of ACE and the shortest action time (ACE 0.6% 30) has been selected to subsequently evaluate the effects of pre-treatment of animals with the probiotic commercial. However, based on the visceral sensitivity results and taking into account the trend towards an increase in mast cells at the highest concentration of ACE, the ACE4% 60 protocol would also be interesting. In any case, it would be convenient to explain in the text in more detail the choice of protocol (lines 334-342).
The discussion is too long and can be difficult to follow. It is recommended to synthesize it, focusing on discussing the main results shown in the work, without going too deep into other more speculative aspects. For example, although it is important to briefly discuss the endogenous mechanisms of pain inhibition, the detailed discussion of this topic in the current version might be excessive. This information would be very valuable if the mechanisms involved in the inhibition of visceral pain in this model had been identified experimentally.
Similarly, the discussion of mast cells (associated in the literature with visceral hypersensitivity) is somewhat confusing. Perhaps it would be convenient to focus more on the only protocol that significantly increases its number (ACE 4% 30) and try to correlate it with the absence of inhibition of visceral pain in that case.
Unfortunatelly, the composition of the commercial probiotic used does not show quantitative data for some of its additional components, such as vitamins or minerals. Specifically, it would be interesting to know the amount of vitamin D. In the last paragraph of the discussion, based on bibliography, it is suggested that this vitamin could also influence the results. However, this is a minor issue that probably cannot be verified. On the other hand, experimental animals are not expected to be deficient in this vitamin.
Specific comments
- Line 20: To clarify, it would be convenient to replace "decrease the response" with "decrease the nociceptive response", or something similar.
- Lines 24 and 25. Specify more clearly, modifying the sentence, that the results of the work do not show an activation of the early inhibitory mechanisms after colonic irritation.
- Line 250: SS60 should be replaced by SAL60
- Line 270: Figure 3B. It would be convenient to use the same abbreviations that are used throughout the work: SAL, ACE 4% and ACE 0.6% and add the times 30 or 60.
- Line 290: Figure 4B. It would be convenient to use the same abbreviations that are used throughout the work: SAL, ACE 4% and ACE 0.6% and add the times 30 or 60.
- Lines 308 and 309: It would be convenient to use the same abbreviations that are used throughout the work: SAL, ACE 4% and ACE 0.6% and add the times 30 or 60.
- Line 331: It would be convenient to use the same abbreviations that are used throughout the work: SAL, ACE 4% and ACE 0.6% and add the times 30 or 60.
- Line 336: SS should be replaced by SAL
- Line 590: It would be convenient to add the volume used: 0.7 mL of 0.6% acetic acid
- Line 677: it would be convenient to delete the term of inflammation. The histological results do not seem to show a fully consolidated inflammatory process, due to the short action time of the irritant.
- Line 726: it would be convenient to replace: “these results show…” with another phrase such as, “these results could be explained through a possible early activation of…” or something similar
Author Response
Dear reviewer,
Thank you very much for your help in this reviewing process. We have addressed all the issues raised by the reviewer (please see attached document) and hope the new version of our manuscript is now acceptable for publication.
Kind regards,
Dr Raquel Abalo (on behalf of all co-authors)

Reviewer 2 Report
- Verification of probiotics is excellent, but it is not more convincing to verify prebiotics because there is a possibility that enzymes produced by intestinal bacteria may affect the living body. mosquito?
- There is no problem because it is ethically considered excellently.
- Since the statistical examination does not show a clear P value, I would like to confirm not only "P <0.05" but also the actual P value.
- It is not clear where and where the significant difference is found because there is no asterisk that indicates a significant difference in the figure as a whole.
Author Response

(The authors gave the same response as above.)

Reviewer 3 Report
In this original research, the authors used acetic acid (different concentrations) in rats vs saline solution and studied their effects on visceral sensitivity, colonic mucosal damage and various types of inflammatory cells. In the Phase II of their research, they compared the effects obtained after previous administration of a commercial probiotic (Actimel). The authors are experienced on this topic. I appreciate the clarity of the experiments and the attention to details. The eleven figures are generally well explained and illustrative. Paragraph Discussion is nicely conceived, maybe a bit long, but I would not suggest to cut anything, as everything has its place. Good manuscript. I have some minor questions/comments (below).
- I wonder why the authors chose Actimel, as none of the probiotics included in Actimel is recommended in humans with IBD or IBS, by the guidelines of the respective scientific societies.
- Abstract: In IBD, the current goals do not refer to only alleviating symptoms, but to reaching mucosal healing (in UC) and, eventually, transmural healing (in CD), plus other targets (Ref. STRIDE II). Controlling symptoms is not anymore the goal, please revise. (Turner D, Ricciuto A, Lewis A, D'Amico F, Dhaliwal J, Griffiths AM, Bettenworth D, Sandborn WJ, Sands BE, Reinisch W, Schölmerich J, Bemelman W, Danese S, Mary JY, Rubin D, Colombel JF, Peyrin-Biroulet L, Dotan I, Abreu MT, Dignass A; International Organization for the Study of IBD. STRIDE-II: An Update on the Selecting Therapeutic Targets in Inflammatory Bowel Disease (STRIDE) Initiative of the International Organization for the Study of IBD (IOIBD): Determining Therapeutic Goals for Treat-to-Target strategies in IBD. Gastroenterology. 2021 Apr;160(5):1570-1583). Also, please insert here that the study was performed on rats and there were 72, divided in groups. Please insert the period the study was performed and the location (also in the main text).
- Introduction: A. same for L 46-47 – STRIDE II; controlling symptoms is not anymore the goal. B. L 49-51: Reference 14 should be the ISAPP one (Hill C, et al, 2014) – and the respective definition should be used in the text.
- Materials and Methods: a. Animals – please explain why you chose only male. b. Figure 1 appears clear and well designed. One question please: Why in “B” the acetic acid appears only in green? In “A”, green means 0.6% concentration, while red was used for 4% concentration. Please explain. Later on, I got my answer in section “3.2. PHASE II: Evaluation of the effect of a commercial probiotic.” Maybe it would not hurt to insert a sentence in Figure legend nr. 1 to explain which group was used for the Phase II, so that appears clearer. c. Statistics: Please define SEM before abbreviation (standard error of the mean).
- Results: Fig. 3 B – I would suggest to enlarge the images, as the respective alterations are meaningful.
- Discussion: a. I suggest to emphasize at the beginning only the findings that were statistically significantly different. b. L 658-661: The following sentence refers to a study in mice - ref [46]. The authors wrote “Importantly in the context of our research, positive effects of the administration of probiotics have been observed in the gastrointestinal tract, such as the reduction of inflammation, visceral hypersensitivity and abdominal pain, as well as improvements in the quality of life of patients.” Please explain the insertion here of “quality of life of patients”. c. L 662-663: The authors wrote: “…probiotic is the Bifidobacterium infantis 35624 strain [47]. In fact, it is the strain present in the only approved probiotic drug, to date, for the treatment of IBS (Alflorex®). However, according to the most recent Guidelines regarding probiotics by AGA (AGA Clinical Practice Guidelines on the Role of Probiotics in the Management of Gastrointestinal Disorders by Grace L. Su, et al, published in Gastroenterology, in 2022), it is stated: “In symptomatic children and adults with irritable bowel syndrome, we recommend the use of probiotics only in the context of a clinical trial. d. Line 666 – I suggest removing the term VSL#3, given the recent issues across the world and use the term “De Simone formula”. Strange that the authors were not aware of this problem. e. I also strongly suggest the authors to use the new names of the probiotics from the genus Lactobacillus. The paper was published in 2020! (Zheng et al. Int. J. Syst. Evol. Microbiol. 2020). f. References 44 and 45 – more recent data are available, from systematic reviews, meta-analyses, also from guidelines and they should be preferred. g. I appreciate that the authors mentioned that some findings could be the effect of the other components of Actimel. However, the authors did not write any limitations/weaknesses of their study. Please insert. H. What directions of research do the authors suggest for the future?
- References: many of them are old, while recent references exist. Please update.
Author Response

(The authors gave the same response as above.)
